# Regular rhythmic and audio-visual stimulations enhance procedural learning of a perceptual-motor sequence in healthy adults: A pilot study

Yannick Lagarrigue[1]*, Céline Cappe[2], Jessica Tallet[1]

**1** ToNIC, Toulouse NeuroImaging Center, Université de Toulouse, Inserm, UPS, Toulouse, France, **2** Cerco, Centre de Recherche Cerveau et Cognition, Université de Toulouse, CNRS, UMR 5549, Toulouse, France

* yannicklagarrigue@outlook.fr

**Data Availability Statement:** All relevant data are within the paper and its Supporting information files.

## Abstract

Procedural learning is essential for the effortless execution of many everyday life activities. However, little is known about the conditions influencing the acquisition of procedural skills. The literature suggests that sensory environment may influence the acquisition of perceptual-motor sequences, as tested by a Serial Reaction Time Task. In the current study, we investigated the effects of auditory stimulations on procedural learning of a visuo-motor sequence. Given that the literature shows that regular rhythmic auditory rhythm and multisensory stimulations improve motor speed, we expected to improve procedural learning (reaction times and errors) with repeated practice with auditory stimulations presented either simultaneously with visual stimulations or with a regular tempo, compared to control conditions (e.g., with irregular tempo). Our results suggest that both congruent audio-visual stimulations and regular rhythmic auditory stimulations promote procedural perceptual-motor learning. On the contrary, auditory stimulations with irregular or very quick tempo alter learning. We discuss how regular rhythmic multisensory stimulations may improve procedural learning with respect of a multisensory rhythmic integration process.

## Introduction

Procedural learning refers to the acquisition and retention of motor and cognitive skills with repeated practice [1–5]. It is essential for many everyday life activities such as driving a car or playing a musical instrument but also for reading or writing [6,7]. Many studies showed that repeated practice of a structured perceptual-motor sequence specified by a stimulation-response association improves the speed of the motor responses and leads to the acquisition of the perceptual-motor sequence [8–12].

The Serial Reaction Time Task (SRTT), previously developed by Nissen and Bullemer (1987) [13], has been widely used to study the implicit procedural learning of a visuo-motor sequence in clinical and nonclinical populations [14–17]. As described by [18], in the classic form of the task, participants have to respond as fast and as accurately as possible by pressing

**Funding:** The authors received no specific funding for this work.

**Competing interests:** The authors have declared that no competing interests exist.

one of four keys corresponding to one of four locations of a visual cue. Unbeknownst to the participant, the first blocks of practice are composed of a repeated structured sequence (e.g., 10 items) that is implicitly learned. Then, on the sixth block of practice the sequence is unexpectedly removed and replaced with random trials. Participants who learned the perceptual-motor sequence will respond less quickly and/or less accurately in this random block. Thus, as explained by Robertson (page 10073), "the difference between sequential and random response times provides a specific and sensitive measure of skill acquisition in the SRTT". Indeed, as opposed to the general learning that includes both familiarization and sequence learning, this difference in RT (or errors) reflects the learning of the specific sequence and is called specific learning. As stated by [19], multisensory-training protocols could enhance learning and better approximate natural multisensory settings. Several studies have tested how the manipulation of the perceptual stimulations could improve motor procedural learning. Globally, it appears that stimulations' features play an essential role for learning of a motor sequence [for review see 20,21]. Manipulations refer to stimulation modality [22–24], stimulation type [17,25,26], stimulation-response mapping [27,28] or response effect mapping [29]. Results generally support the role of visuo-spatial stimulations to memorize the sequence. The effects of auditory stimulations on learning of a visuo-motor sequence have been more rarely studied. Yet, several studies suggest that the temporal regularity of the sequence is learned concomitantly with the visuo-spatial sequence itself [30–32]. More precisely, [32] showed that temporal patterns can be learned when the intervals are associated with concrete events, such as specific visual stimuli or finger movements, and that temporal and spatial parameters are learnt in an integrated fashion allowing to acquire the order of a repeated sequence. Given that temporal regularity detection is best accomplished with the auditory system [33,34], we hypothesized that providing auditory stimulations could facilitate the detection of the temporal regularity of the perceptual-motor sequence and so facilitate its learning.

Two types of mechanisms have been discovered to explain the benefits of auditory stimulations on motor control and learning: multisensory integration and audio-motor entrainment. Firstly, auditory stimulations provided simultaneously with visual stimulations can lead to *multisensory integration*. [35] showed that some neural cells of animals have higher neuronal activity in response to multisensory stimulations compared to unisensory stimulations. To be integrated as a coherent whole, stimulations have to be presented in a spatial and temporal "binding window" [i.e., 36–39]. Multisensory integration leads to behavioral improvements, i.e. a reduction of simple reaction time [40] and choice reaction times [41] and a quickening of the detection of visual targets [e.g. 42]. Multisensory stimulations seem also to benefit perceptual learning [19]. Particularly, [43] found faster improvements on a motion-detection learning task with multisensory stimulations compared with unisensory stimulations. On this basis, it is possible that multisensory stimulations could also enhance perceptual-motor learning.

Secondly, another way to promote procedural perceptual-motor learning with auditory stimulations comes from studies on Regular Auditory Stimulations (RegAud). RegAud have been proved to induce benefits on motor control [e.g., 44]. In many situations, movements are spontaneously attracted to external regular rhythms although participants are not instructed to synchronize with [see review of 45,46]. RegAud induce a priming effect which can lead to a facilitation to produce voluntary movements called audio-motor entrainment [44,47]. This facilitation consists in an improvement of stability of movements in both time and space [48–50] and these effects are still observable even when attention is focused on the visual modality [51]. The spontaneous sensorimotor synchronization with an auditory rhythm can be explained by the involvement of motor cerebral areas, particularly the supplementary motor area and the primary motor cortex, in rhythm perception and production [52–55]. Moreover, listening of auditory stimulations with regular tempo, such as a metronome, modulates

corticospinal excitability measured via motor-evoked potentials (measured thanks to TMS) [56] and creates a stable time scale with a predictable pace to which the motor system adjusts for motor programming [54,57]. [59] showed that audio-motor synchronization is more accurate with simple metrics (regular intervals) than with irregular metrics (i.e., irregular intervals). Benefits of RegAud on motor control could be explained by combined activations of both auditory and motor areas [57–60]. It remains to explore the possible benefits of RegAud on perceptual-motor learning. Previous studies suggested that the temporal organization of the stimulations is an essential part of the perceptual-motor sequence learning [30,32,61]. Hence, the predictable tempo of the regular auditory stimulations may modulate the possible improvement of procedural learning. Thus, we hypothesize that RegAud improve procedural learning of a perceptual-motor sequence.

On these bases, the aim of our study is to investigate the possible effects of auditory stimulations on procedural learning evaluated with a SRTT. Auditory stimulations are provided either in congruency with visual stimulations (Congruent Audio-Visual, CongrAV) or regularly (RegAud). Possible effects of these additional auditory stimulations are controlled with four conditions: visual only stimulations (Visual Only, VisOnly), incongruent audio-visual stimulations (Non-Congruent Audio-Visual, NonCongrAV), and non-regular stimulations (Irregular Auditory Stimulations, IrregAud). Our main hypothesis is that auditory stimulations presented congruently with visual stimulations (CongrAV condition) and regular rhythmic auditory stimulations (RegAud condition) will enhance procedural learning compared to control conditions. We also hypothesized that the effects of the regular auditory stimulations (RegAud) could be linked to their speed and if this speed is not suitable to the task, we would not observe any benefits. Thus, speed effects are controlled with a condition with Quick Regular Auditory Stimulations (FastRhyth).

## Method

### Participants

Sixty right-handed (laterality quotient = 77.59 ± 21.45) adults participated to the study (32 females). Participants were undergraduate students pursuing sports science courses in Toulouse university. They were from 18 to 30 years old (mean age = 21.80 ± 2.53), reported normal or corrected-to-normal vision and hearing and were naïve as to the purpose of the study. They did not practice music more than two hours per week during a maximum of two years. They were randomly and equally assigned to the six different conditions. The study was conducted in accordance with the Declaration of Helsinki and approved by the Inserm (Institut National de la Santé et de la Recherche Médicale) ethical committee (Institutional Review Board IRB00003888—agreement n˚14–156). Before the experiment beginning, all volunteers provided a verbal informed consent and documented a written form specifying their motivation to participate to the study.

### Materials

Stimulation presentation and data collection were achieved using experimental software Presentation version 17.2 (Neurobehavioral System Inc, Albany, CA) which provide a precision under 1 millisecond for motor response measures [62]. The laptop was connected to an external display (40cm, refreshing at 60Hz) and to an adapted keyboard. On this keyboard, the keys were removed except those corresponding to the four letters D, F, G and H which were marked with yellow pallets.

Preparatory attention and divided attention were tested with the standardized alert phasic test and divided attention test of the Test of Attentional Performance (TAP, Version 2.3 Zimmermann and Fimm, 2002).

## Procedure

The participant was seated in standardized sitting posture in a quiet room without visual or auditory distractors. The viewing distance was approximately 80cm and the keyboard was at 50cm from the screen. The experimenter space was separated from the participant space by a curtain.

Before the experiment, useful data about the participants (date of birth, gender, handedness assessed with the Edinburgh Handedness Inventory, [63]), were collected.

**Test of Attentional Performance (TAP).** Attentional performance was assessed to explore the link between visuo-motor learning and attentional skills. This measure was also done to make sure that groups did not differ in terms of attentional functions. Each of the two neuropsychological tests was composed of two parts: a phase to familiarize the participant with the instructions and a test phase in which the results were recorded. We assessed two attentional functions: preparatory attention (alert phasic test) and divided attention. Note that one participant (RegAud condition) did not perform the attentional tests for time issues.

**The Serial Reaction Time Task (SRTT).** We used a version of the serial reaction time task (SRTT) in which participants were instructed to respond using the four fingers except thumbs (index, middle, ring, and little finger) of their right hand to press the D, F, G, and H keys of the keyboard. Each of the four possible keys corresponded to one of the four stimulation locations. The four possible stimulation positions were specified by four equally spaced gray boxes, each a 2cm square, presented on a computer screen so that the stimulation-response mapping would be compatible with the keyboard. On each stimulation-response mapping, one of the four boxes on the monitor was colored in yellow, and the participant's task was to press the corresponding key on the keyboard (Fig 1) as fast as possible ("Try to go as fast as possible and make as less as possible mistakes"). Once a key was pressed, the computer recorded the participant's reaction time and then moved to a different box with an interval of 250ms before the next target. If the participant did not press any key, the stimulation remained for 3000ms on the screen and then went on with the next stimulus.

All participants went through 7 Blocks containing 100 items:

- 1 Block of familiarization (B0) displaying a sequence of 10 positions repeated 10 times (100 items). It was performed in the same condition as the following blocks in order and aimed to make sure that there were no significant differences between the groups in performance at the beginning of the experiment (B0 can be considered as a baseline),

- Then, 5 Blocks of practice of a same sequence displaying a repeated pattern of 10 positions presented ten times (100 items each block, *i.e.*, 500 items in total) (B1 to B5) in order to test general learning.

- Finally, a last Block (B6) presented the visual stimulations in a pseudo-random fashion (100 items). The sequence with the repeating pattern of positions was no longer played out. This Block aimed to test specific learning of the sequence.

We used several different sequences instead of a unique one because learning can depend on the sequence used [64] and it opens up the possibility that the obtained results may be specific to a particular fixed sequence. Thus, each group was we proposed different controlled sequences sharing the same rules:

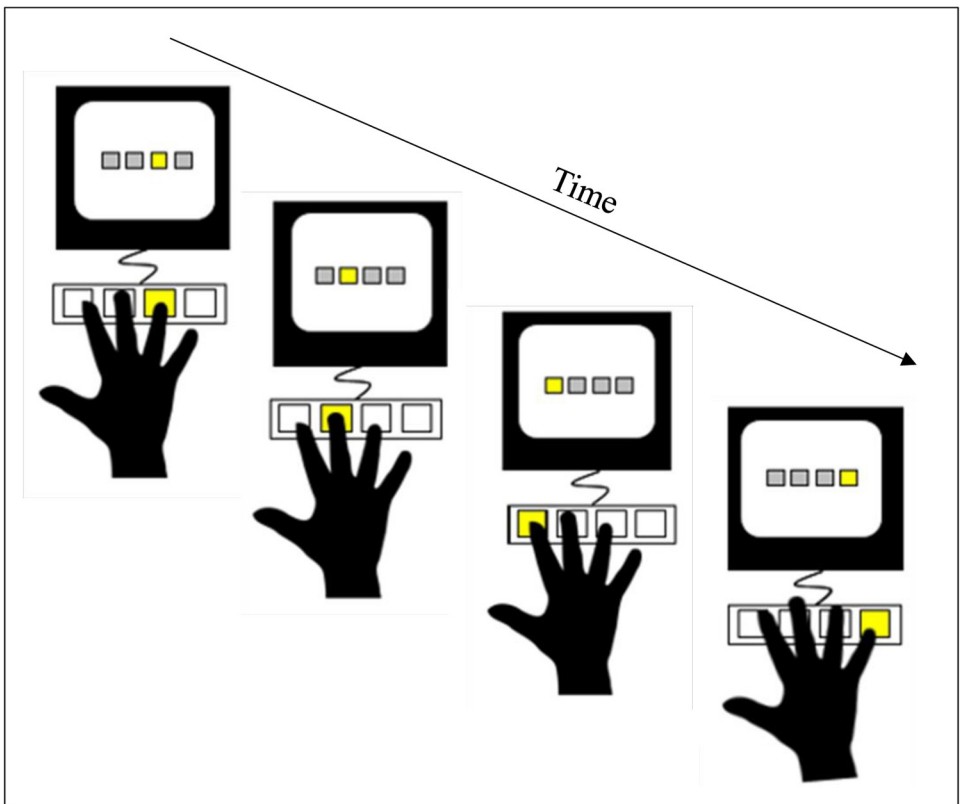

**Fig 1. Serial reaction time task.** Each finger is associated with a response key and each key is associated with a visual cue. When a box lighted the participant had to press the corresponding key as quickly as possible.

- The same position could not appear on successive trials

- Each position appeared an equal number of times

- The sequence could not contain runs (e.g., 1234)

- The sequence could not contain trills of four units (e.g., 1313).

On this basis, four sequences of ten positions were generated (sequence A: *1 3 4 2 3 1 4 2 1 4*, sequence B: *2 4 1 3 4 2 1 4 3 1*, sequence C: *3 1 4 2 1 3 4 1 2 4*, sequence D: *4 2 3 1 2 4 1 3 4 1*). Each sequence was attributed in a counterbalanced manner to the participants for each condition. The selected sequence of B0 was different from this selected for B1 to B5 in order to avoid a possible transfer of learning between two specific sequences. For Block 6 (B6), all participants performed the same pseudo-random stimulations following the previous rules applied to 100 items (3,2,1,3,4,1,3,4,2,3,1,2,4,3,1,2,3,4,2,3,1,4,3,2, 4,1,2,4,1,3,2,4,1,2,4,1,3,4,2,1,4,2,3,1,4,3,1,2,4,3,1,2,4,3,1,2,4,3,1,2,4,2,3,1,4,1,3,2,1,3,4,2,1- ,3,2,1,3,2,1,4,2,3,1,4,3,2,4,1,2,4,1,3,4,2,3,4,1,4,2,3).

All participants were randomly and equally assigned to one of the six different conditions.

- In the Visual Only (VisOnly) condition, visual stimulations were presented without auditory stimulations.

- In the Congruent Audio-Visual (CongrAV) condition, an auditory stimulation was presented at exactly the same time as each visual cue.

- In the Non-Congruent Audio-Visual (NonCongrAV) condition, an auditory stimulation was presented 200ms after each visual cue. If the participants pressed a key before this delay, the auditory stimulation was not presented.

- In the Regular Rhythmic Auditory Stimulations (RegAud) condition, auditory stimulations were presented every 500ms independently of visual stimulations and participants' responses.

- In the Irregular Rhythmic Auditory Stimulations (IrregAud) condition, auditory stimulations were presented irregularly and independently of visual stimulations and participants' responses. There was the same number of auditory stimulations in this soundtrack than in the RegAud condition but intervals were pseudo-randomly generated in a range of 0.022s to 2.891s and a mean of 0.494s. This pattern was programmed with the free software Audacity. The same soundtrack was presented in all blocks.

- In the Quick Regular Rhythmic Auditory Stimulations (FastRhyth) condition, auditory stimulations were presented every 300ms independently of visual stimulations and participants' responses.

In CongrAV, NonCongrAV, RegAud, IrregAud and FastRhyth conditions the auditory stimuli were presented via two external speakers placed on both sides of the screen and consisted in a 500Hz, 100ms sinewave presented at 80dB. Participants were told that there would be auditory stimuli but we did not tell them anything about their purpose. In the conditions CongrAV, NonCongrAV, RegAud, IrregAud and FastRhyth, the auditory stimuli were presented from Block 0 to Block 6. Note that B0 is performed in the same condition as the following Blocks because changing the way how stimuli are introduced between B0 and B1 would have possibly induced confounding effects on participants' performance, hence confusing the interpretation of the results of the general learning phase. Moreover, a removal of the auditory stimuli in the random block (B6) would have induce a double change (sequence and auditory stimuli) for participants and it would have been difficult to determine if decrease change in performance would be due to a change in the sequence or the removal of the auditory stimuli.

The order of the SRTT and TAP tests was counter-balanced for each participant to prevent training or fatigue effects. At the end of the random Block (B6), the participants' explicit knowledge of the sequence was measured by asking them whether or not they noticed a repeated sequence.

The entire experiment took approximately 1h.

## Data analyses

For all analyses, incorrect responses were not included in the RT analyses.

**Pre-tests.** The laterality quotient was assessed with the Edinburgh Handedness Inventory [63]. For the two attentional tests, mean reaction times and errors were computed by the TAP software. For the B0, the averaged reaction times (RTmean), variability of reaction times (RTsd) and errors were computed and compared between Conditions.

**SRTT.** Average reaction times (RTmean), variability of reaction times (RTsd) and errors for all participants in each Condition were computed across trials of each of the block. Difference in performance between Blocks 1 to 5 was considered as a measure of *general learning* (difference B1-B5 = RTmeanB1-B5, RTsdB1-B5, ErrorB1-B5) whereas performance in Blocks 5 to 6 (fixed to pseudorandomized order of visual stimulations) was considered as a measure of *specific learning* of the sequence (difference B6-B5 = RTmeanB6-B5, RTsdB6-B5, ErrorB6-B5). A decrease in RTmean, RTsd and errors was expected between the B1 and B5 to

give evidence of general learning and an increase in RTmean, RTsd and errors is expected between B5 and B6 (fixed to randomized order of visual stimulations) to give evidence of specific sequence learning.

**Explicit knowledge.** Three kinds of responses were recorded to answer the question "Did you notice that the presentation of the boxes followed a repeated sequence?": yes, no and maybe. We then computed the percentage of yes in each Condition.

## Statistical analyses

The equality of variances was assessed by Levene's test and normality of the distribution was tested by the Kolmogorov-Smirnov test. If assumptions were verified (at least 50% of data with equality of variance and normal distribution), analyses of variance (ANOVAs) were used. If data did not satisfy the criteria of equality of variance or normality, nonparametric tests (Friedman, Mann-Whitney and Wilcoxon signed rank test) were used. When appropriate, the data were further analyzed with post-hoc analysis (Fischer test).

A significance level of 0.05 was adopted for all analyses. Only significant results ($p < .05$) are reported in the Results section. All results are plotted using means and standard errors.

**Pre-tests.** To make sure that laterality of participants did not differ between Conditions, one-way ANOVAs with Conditions as a Factor was conducted on the Laterality Quotient.

To make sure that attentional characteristics of participants did not differ between Conditions prior to the SRTT, one-way ANOVAs with Conditions as a Factor were conducted on the mean reaction times and errors for the two attentional tests. Pearson's correlations were used to explore the link between attentional performance (alert phasic and divided attention) and the SRTT general learning score (difference B1-B5 = RTmeanB1-B5, RTsdB1-B5, ErrorB6-B5) and specific learning score (difference B6-B5 = RTmeanB6-B5, RTsdB6-B5, ErrorB6-B5).

**B0 analyses.** To make sure that participants did not differ between Conditions prior to the SRTT, one-way ANOVAs with Conditions as a Factor were conducted on RTmean, RTsd and errors. Pearson's correlations were used to explore the link between B0 performance (RTmean, RTsd and errors) and the SRTT general learning score (difference B1-B5 = RTmeanB1-B5, RTsdB1-B5, ErrorB6-B5) and specific learning score (difference B6-B5 = RTmeanB6-B5, RTsdB6-B5, ErrorB6-B5).

**SRTT.** To determine general learning (B1-B5) for all conditions, Conditions x Blocks ANOVAs with Conditions (VisOnly, CongrAV, NonCongrAV, RegAud, FastRhyth and IrregAud) as between-subject factor and Blocks (B1 to B5) as a repeated measure were conducted on RTmean. One-way ANOVA was conducted on the RTmean differences B1-B5 (RTmeanB1-B5) to compare general learning evolution between Conditions. Friedman tests were used to assess RTsd and errors' evolution in each Conditions.

To determine sequence-specific learning (B5-B6), Conditions x Blocks ANOVAs with Conditions (VisOnly, CongrAV, NonCongrAV, RegAud, FastRhyth and IrregAud) as between-subject factor and Blocks (B5 to B6) as a repeated measure were conducted on RTmean and errors from B5 to B6. One-way ANOVAs were conducted on the RTmean and errors differences B6-B5 (RTmeanB6-B5 and ErrorB6-B5) to compare specific learning evolution between Conditions. Wilcoxon signed rank tests were used to compare RTsd evolution between Conditions from B5 to B6 (RTsdB6-B5).

We estimated the Bayes Factor for these data using JASP [65]. The Bayes Factor is used to compare two hypotheses (H0 and H1) based on collected data. It tells how much more likely one hypothesis is to be true compared to the other (e.g., [66–69]. The Bayes factor ($BF_{10}$) measures the likelihood of H0 vs. H1 given our data. Although Bayes factors are defined on a

continuous scale, several researchers have proposed to subdivide the scale into discrete evidential categories [70]. We used the standard non-informative Cauchy prior in JASP with a default width of 0.707.

**Explicit knowledge.** We computed the percentage of "yes" reported by participants in each Condition. Kruskal-Wallis ANOVAs by Ranks were used to compare this percentage between Conditions. We also compared the specific learning score of participants who noticed a repeating pattern of position and participants who did not with t.tests on the RTmean and errors differences B6-B5 (RTmeanB6-B5 and ErrorB6-B5).

## Results

### Pre-tests

No difference was found between Conditions on the participants' mean reaction time (RTmean) for the divided attention task and the phasic arousal index. No difference was found on the Laterality Quotient and at B0 on the RTmean, the variation of the reaction time (RTsd) or errors (Table 1a). Moreover, we did not find correlation between the performance at the first block of practice (B0) and SRTT learning scores, or between the two scores of the attentional tasks and SRTT learning scores (see Table 1b).

**Table 1.** a: Participants' results for the Block 0 (B0) and the Tests of Attentional Performance (TAP). b: Pearson correlations' results between Block 0 (B0), the Tests of Attentional Performance (TAP) and SRTT learning scores.

| | CONDITIONS | | | | | | F and p values |
|---|---|---|---|---|---|---|---|
| | VisOnly | CongrAV | NonCongrAV | RegAud | IrregAud | FastRhyth | - |
| Number of participants | 10 | 10 | 10 | 10 | 10 | 10 | - |
| Laterality Quotient | 72.00 (±19.85) | 79.03 (±23.72) | 72.85 (±27.88) | 75.06 (±16.46) | 80.81 (±25.17) | 85.76 (±14.95) | $F_{(5, 54)} = 0.6, p = .712$ |
| B0 RTmean | 409.70 (±56.28) | 453.16 (±80.82) | 452.52 (±65.27) | 398.83 (±44.98) | 440.15 (±80.20) | 428.95 (±54.09) | $F_{(5, 54)} = 1.2, p = .323$ |
| B0 RTsd | 113.24 (±37.39) | 108.77 (±30.26) | 136.79 (±53.17) | 100.42 (±31.07) | 122.99 (±55.39) | 120.74 (±64.48) | $F_{(5, 54)} = 0.7, p = .613$ |
| B0 errors | 2.5 (±1.90) | 4.6 (±4.40) | 2.0 (±2.87) | 5.4 (±3.03) | 3.2 (±2.10) | 3.8 (±2.40) | $F_{(5, 54)} = 2.0, p = .101$ |
| Phasic Alert (PA) | 0.311 (±0.05) | 0.022 (±0.06) | 0.018 (±0.06) | 0.012 (±0.07) | -0.001 (±0.04) | 0.057 (±0.08) | $F_{(5, 53)} = 1.0, p = .417$ |
| Divided Attention (DA) | 596.76 (±51.66) | 661.28 (±90.35) | 632.88 (±68.39) | 599.25 (±68.39) | 641.06 (±77.62) | 619.00 (±61.84) | $F_{(5, 53)} = 1.3, p = .274$ |

| | B0 RTMEAN | B0 RTSD | B0 ERRORS | PHASIC ALERT (PA) | DIVIDED ATTENTION (DA) |
|---|---|---|---|---|---|
| **RTMEAN B1-B5** | r = .022<br>p = .865 | r = -.135<br>p = .305 | r = -.032<br>p = .810 | r < .001<br>p = .995 | r = -.139<br>p = 295 |
| **RTSD B1-B5** | r = .046<br>p = .726 | r = -.018<br>p = .893 | r = .090<br>p = .495 | r = .024<br>p = .857 | r = -.101<br>p = .447 |
| **ERROR B1-B5** | r = -.169<br>p = .197 | r = -.116<br>p = .377 | r = -.003<br>p = .984 | r = .252<br>p = .054 | r = .037<br>p = .784 |
| **RTMEAN B6-B5** | r = .163<br>p = .214 | r = -.032<br>p = .810 | r = .141<br>p = .281 | r = -.066<br>p = .617 | r = -.211<br>p = .110 |
| **RTSD B6-B5** | r = -.145<br>p = .270 | r = .056<br>p = .670 | r = .026<br>p = .841 | r = .106<br>p = .423 | r = -.078<br>p = .560 |
| **ERROR B6-B5** | r = -.238<br>p = .068 | r = -.110<br>p = .407 | r = .249<br>p = .055 | r = .011<br>p = .934 | r = .056<br>p = .672 |

Numbers in parentheses represent standard deviations. RTmean = Reaction Times / RTsd = standard deviation of the reaction time / VisOnly = Visual Only / CongrAV = Congruent Audio-Visual / NonCongrAV = Non-Congruent Audio-Visual / RegAud = Regular Auditory Stimulations / IrregAud = Irregular Auditory Stimulations / FastRhyth = Quick tempo Rhythmic Auditory Stimulations. ANOVAs were used to compare Conditions.
None of the correlations was significant.

## General learning (B1-B5)

ANOVA on RT mean revealed a significant effect of Block ($F$ (4, 216) = 27.11, $p < .001$, $\eta^2_P$ = .334, $BF_{10} > 100$). We found $BF_{10}$ greater than 100 which corresponds to decisive evidence for $H_1$. As illustrated in Fig 2a, RTmean decreased from Blocks 1 to 5. ANOVA also revealed a significant interaction between Blocks and Conditions ($F$ (20, 216) = 1.76, $p = .027$, $\eta^2_P$ = .140, $BF_{10}$ = 1.98) suggesting that the evolution of RTmean differed between Conditions. However, we found $BF_{10}$ to be 1.98 which corresponds to anecdotal evidence in favor of $H_1$.

To explore more precisely this result, we conducted an ANOVA with Conditions as a Factor on RTmeanB1-B5 which revealed a significant Condition effect ($F$ (5, 54) = 2.84, $p = .024$, $\eta^2_P$ = .208, $BF_{10}$ = 2.50). We found $BF_{10}$ to be 2.50 which correspond to anecdotal evidence in favor of $H_1$. Post-hoc Fisher tests revealed that the RTmeanB1-B5 was lower for the FastRhyth Condition than for all the other Conditions (VisOnly: $p = .007$; CongrAV: $p = .014$; NonCongrAV: $p = .003$; RegAud: $p = .005$; IrregAud: $p = .005$).

In order to remove inter-group differences in terms of absolute response speed and allow for more sensitive differences in general learning, each participant's observation on each measure was converted to a z score standardized on participant's mean and variance (pooled across blocks). As previous, ANOVA revealed significant block effect on general learning phase ($F$ (4,236) = 23.67, $p < .001$, $\eta^2_P$ = .286, $BF_{10} > 1000$) (Fig 2b).

For general learning, Friedman tests revealed a significant Block effect only in the IrregAud Condition for both errors ($X^2_R$ (10, 4) = 11.55, $p = .021$) and RTsd ($X^2_R$ (10, 4) = 10.96, $p = .027$), suggesting that errors and RTsd increased in this Condition from B1 to B5 (Fig 3a and 3b respectively).

## Specific learning (B5-B6)

The ANOVA on RTmean from B5 to B6 revealed a significant Block effect ($F$ (1, 54) = 161.14, $p < .001$, $\eta^2_P$ = .749, $BF_{10} > 100$). We found $BF_{10}$ greater than 100 which corresponds to decisive evidence for $H_1$. As illustrated in Fig 2, RTmean increased for all Conditions.

The Block effect from B5 to B6 was also significant on errors ($F$ (1, 54) = 39.75, $p < .001$, $\eta^2_P$ = .424, $BF_{10} > 100$), suggesting that errors increased between B5 and B6 for all Conditions (Fig 4a). We found $BF_{10}$ greater than 100 which corresponds to decisive evidence for $H_1$. Moreover, ANOVA conducted on the ErrorB6-B5 confirmed the significant Condition effect ($F$ (5, 54) = 3.83, $p = .005$, $\eta^2_P$ = .262, $BF_{10}$ = 9.42). We found $BF_{10}$ to be 9.42 which corresponds from substantial to strong evidence for $H_1$. Post-hoc Fisher tests revealed that ErrorB6-B5 was higher in the CongrAV Condition than in the VisOnly Condition ($p = .006$) and FastRhyth ($p = .020$) (Fig 4b). Moreover, ErrorB6-B5 was higher in the RegAud Condition than in the VisOnly ($p < .001$), NonCongrAV ($p = .016$), IrregAud ($p = .049$) and FastRhyth Conditions ($p = .003$) (Fig 4b).

In order to remove inter-group differences in terms of absolute response speed and allow for more sensitive differences in specific learning, each participant's observation on each measure was converted to a z-score standardized on participant's mean and variance (pooled across blocks). As previous, ANOVA revealed significant block effect on specific learning phase ($F$ (1, 59) = 393.21, $p < .001$, $\eta^2_P$ = .870, $BF_{10} > 1000$) (Fig 2b).

## Explicit knowledge

Of the 60 participants, 39 (65%) reported that they had perceived a repeated pattern including 6 participants in the VisOnly condition, 9 in the CongrAV condition, 6 in the NonCongrAV condition, 6 in the RegAud condition, 5 in the IrregAud condition and 7 in the FastRhyth

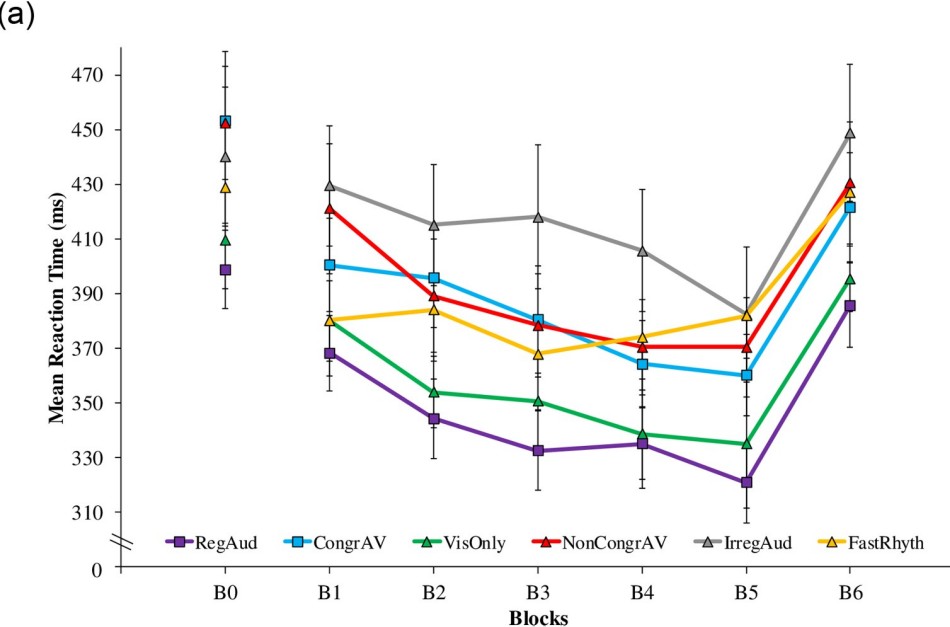

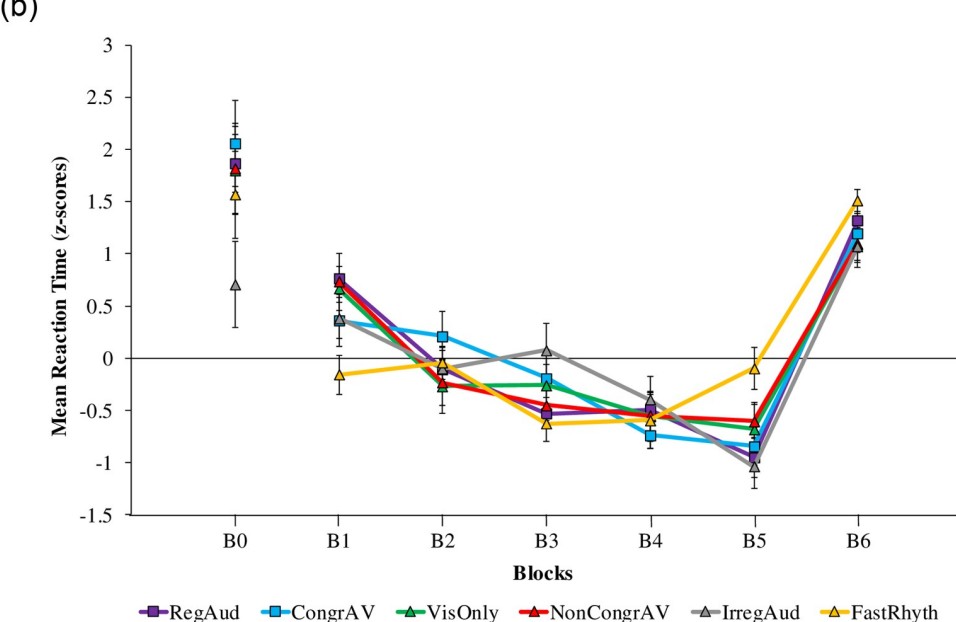

**Fig 2.** a. Mean Reaction Times for general learning (from B1 to B5) and for specific learning (from B5 to B6) of all Conditions. Regular Rhythmic Auditory Stimulations (RegAud in purple squares), Congruent Audio-Visual (CongrAV in blue squares), Visual Only (VisOnly in green triangles), Non-Congruent Audio-Visual (NonCongrAV in red triangles), Irregular Auditory Stimulations (IrregAud in grey triangles) and Quick Rhythmic Auditory Stimulations (FastRhyth in yellow triangles). Vertical bars represent the standard errors. b. Mean Reaction Times (z-scores) for general learning (from B1 to B5) and for specific learning (from B5 to B6) of all Conditions. Regular Rhythmic Auditory Stimulations (RegAud in purple squares), Congruent Audio-Visual (CongrAV in blue squares), Visual Only (VisOnly in green triangles), Non-Congruent Audio-Visual (NonCongrAV in red triangles), Irregular Auditory Stimulations (IrregAud in grey triangles) and Quick Rhythmic Auditory Stimulations (FastRhyth in yellow triangles). Vertical bars represent the standard errors.

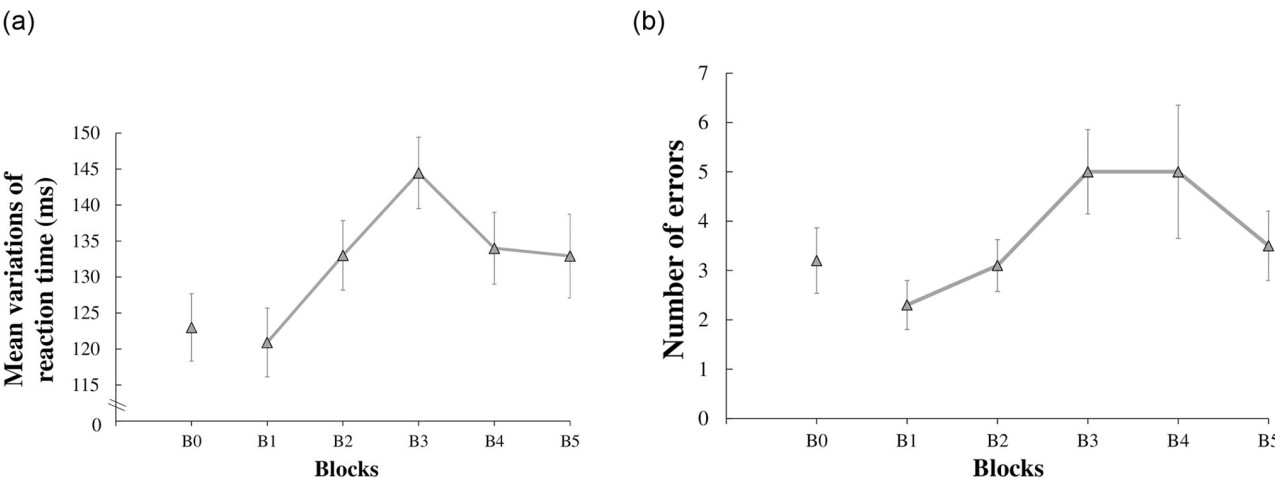

**Fig 3.** a. Mean variations of Reaction Times (RTds) for Irregular Auditory Stimulations Condition (IrregAud) over general learning (B1 to B5). Vertical bars represent the standard error. b. Number of errors for Irregular Auditory Stimulations Condition (IrregAud) over general learning (B1 to B5). Vertical bars represent the standard errors.

Condition. There were no significant differences between Conditions in explicit knowledge ($H$ (5,60) = 4.11, $p$ = .534)).

The t.tests on the RTmean and errors differences B6-B5 (RTmeanB6-B5 and ErrorB6-B5) between participant who noticed a repeating pattern of positions and the one who did not were not significant.

## Discussion

The present study aimed to investigate the effects of auditory stimulations on procedural learning of a visuo-motor sequence. To this aim, auditory stimulations were introduced during a SRTT, either with a regular rhythm (RegAud) or in temporal congruency with visual stimulations (CVA). These conditions were compared to four control conditions: without auditory stimulations (VisOnly), with incongruent audio-visual stimulations (NonCongrAV), with irregular auditory stimulations (IrregAud) or with a quick tempo regular rhythm (FastRhyth). Globally, our results are in accordance with our hypotheses and indicate that both rhythmic auditory stimulations (RegAud) and congruent audio-visual (CongrAV) stimulations enhance procedural learning. This improvement concerns the specific learning, as attested by a larger increase of errors when randomized order of visual stimulations is introduced.

Firstly, it is important to note that these results are not related to laterality, attentional scores or performance at B0, that is, at the beginning of the SRTT. Moreover, all conditions lead to the same explicit detection of the sequence. There is a huge debate in the literature about the link between awareness and what is learned [71,72]. In our study, although some subjects became aware of a repeating pattern during the learning phase (B1 to B5), there were no learning differences between aware and unaware subjects, as also shown by [28]. Hence, the detection of the repeating pattern cannot explain the differences in learning between the conditions. However, further investigations about explicit knowledge associated to learning with and without auditory stimuli are needed, for example with subjective scale with more gradings, which might be more sensitive to differences. Secondly, given that RegAud and CongrAV conditions led to better improvement compared to control conditions with both auditory and visual stimulations, one cannot explain the differences in learning by the addition of two stimulations compared to one cue. Thirdly, attentional profile is not linked to the learning

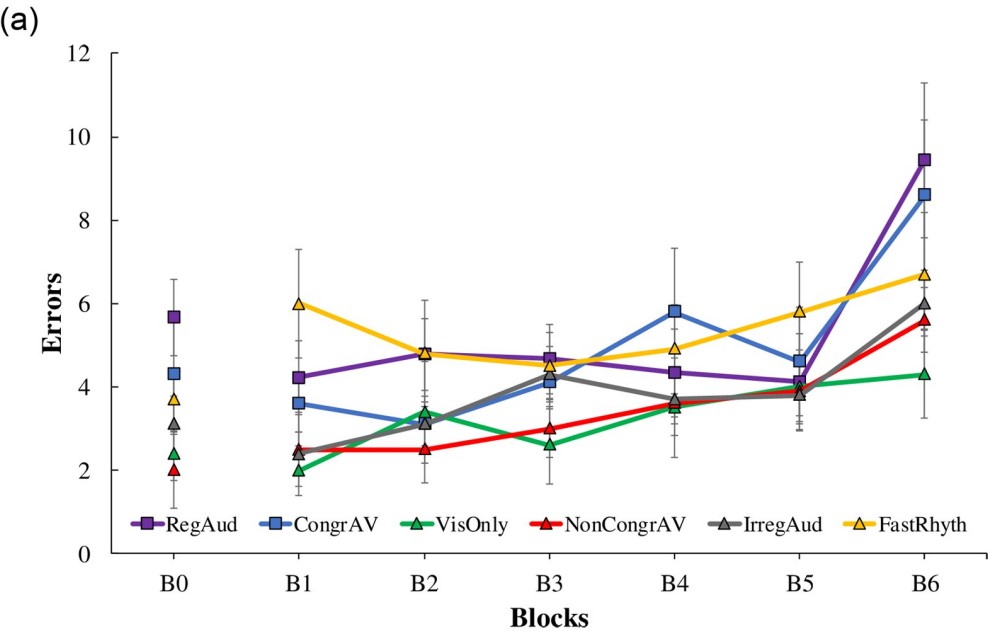

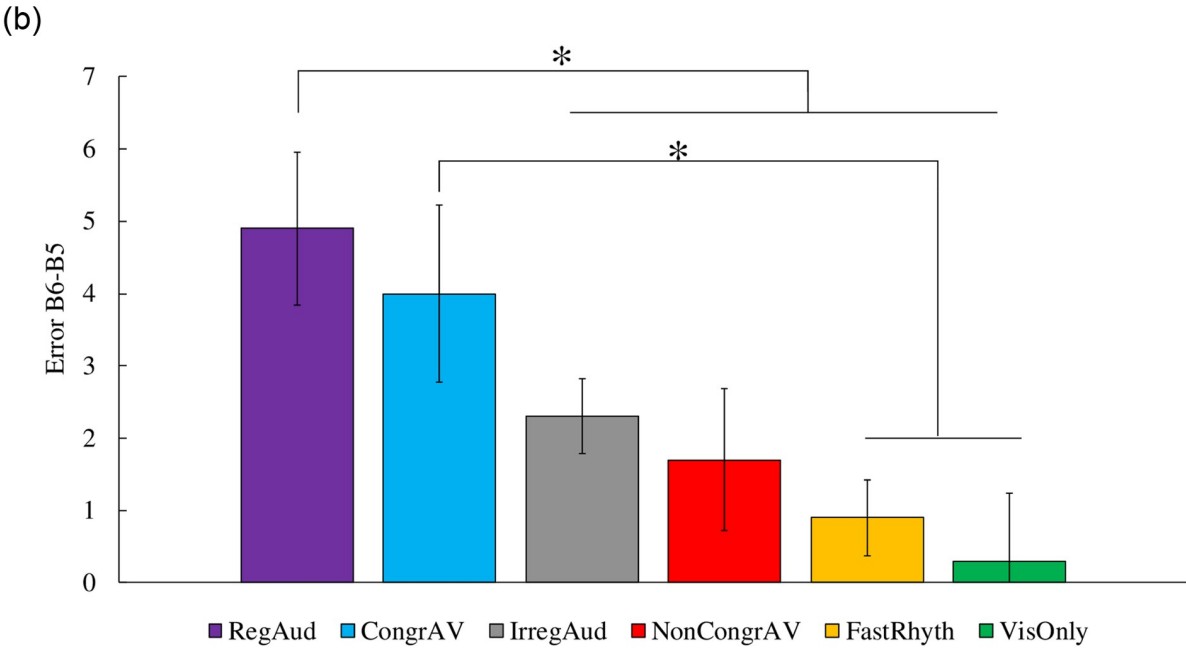

**Fig 4.** a. Errors for general learning (from B1 to B5) and for specific learning (from B5 to B6) of all Conditions. Regular Rhythmic Auditory Stimulations (RegAud in purple), Congruent Audio-Visual stimulations (CongrAV in blue), Irregular Auditory Stimulations (IrregAud in grey), Non-Congruent Audio-Visual stimulations (NonCongrAV in red), Quick tempo Rhythmic Auditory Stimulations (FastRhyth in yellow) and Visual Only (VisOnly in green). Vertical bars represent the standard errors. b. Error differences during specific learning from B5 to B6 (ErrorB6-B5) of all Conditions. Regular Rhythmic Auditory Stimulations (RegAud in purple), Congruent Audio-Visual stimulations (CongrAV in blue), Irregular Auditory Stimulations (IrregAud in grey), Non-Congruent Audio-Visual stimulations (NonCongrAV in red), Quick tempo Rhythmic Auditory Stimulations (FastRhyth in yellow) and Visual Only (VisOnly in green). Vertical bars represent the standard errors.

process. It is in line with several studies showing no links between different executive function tasks and implicit learning [73,74]. Thus, the way by which the auditory stimulations are delivered is not likely to be responsible for the benefits. We discuss the learning improvement in the RegAud and the CongrAV conditions with respect of the involvement of a multisensory rhythmic integration process.

## Specific learning can occur without general learning

In two conditions (IrregAud and FastRhyth) we found specific learning without general learning. It is in accordance with previous findings showing that it is possible to enhance general learning but not sequence-specific learning [see 75]. Thus, our results support the idea that general and specific learning are two distinct processes which are subserved by distinct neural correlates [76]. The first one corresponds to stimulus-based mappings and the second one to internalized sequence representation, or response-based mappings [e.g., 76,77].

## General learning is lower with irregular stimulations and quick tempo auditory stimulations

Errors are discarded from a large number of studies using SRTT although they are important indicators of learning. Indeed, SRTT involves a permanent speed–accuracy trade-off and learning can be attested by a concomitant decrease in RTmean and/or errors. If, an increase in errors is associated with a decrease in RTmean or a decrease in RTmean is associated with an increase in errors, it is not clear whether this improvement reflects a learning effect or only a change of strategy. However, if one of these two variables decreases while the other one remains stable, it means that the performance increase. Interestingly [78], showed that performance of accuracy and speed depend on the instructions (more directed to speed or accuracy) but the learning effect occurs in both cases. Given that our results show differences in conditions on errors only, it is possible that our instructions were more emphasized on accuracy than speed.

Taken both variables into account, our results reveal that the IrregAud condition did not lead to a general learning because it induced a decrease in RTmean concomitant to an increase in errors between B1 and B5. Due to the irregularity (non-isochrony) of the auditory stimulations in the IrregAud condition, it is likely that extracting a temporal pattern of irregular auditory stimulations is more difficult than regular rhythmic auditory stimulations [34,52,79–81]. During a motor task, the introduction of irrelevant auditory stimulation negatively influences motor control [82]. Indeed, [37] showed that irrelevant sounds can cause a disengagement of attention from the task. In this case, participants attempt to suppress the distractors in order to complete successfully the motor task. Other studies showed that auditory distraction alters both visual attention and motor control [83–85]. Hence, introducing irregular auditory stimulations could have generated a distractor effect which limits the attentional focus on the SRTT and could have limited the general learning. Our results also highlight a higher RT variability with the irregular auditory stimulations compared to the other conditions. Again, this is consistent with results of studies showing that providing rhythmic auditory stimulations automatically attracts the tempo of tapping and leads to better stability of movement tempo compared to control conditions without auditory stimulations. Particularly, [86] suggested that auditory rhythms can modify parameters related to the motor production, especially by reducing the variability of muscle activity during the preparatory period.

The decrease in RTmean was lower in the quick tempo rhythmic auditory stimulations (FastRhyth) condition, suggesting that general learning was lower in this condition compared to the other conditions. Given that the only difference between FastRhyth and RegAud is the speed of the auditory stimulations' tempo, it suggests that this speed affects motor learning.

The delay between a participant's response and the next stimulation was 200ms. Thus, with a tempo at 300ms when a motor response occurred at the same time as an auditory stimulation, the next auditory stimulation occurred 100ms after the next visual stimulation, which is too short for the participant to respond. Indeed, the RTmean mean achieved at the last Block (B5) in the FastRhyth condition was 381,93 ($\pm$ 61,13ms). Literature shows that audio-motor entrainment is strongest when the tempo of the external rhythm is close to the spontaneous movement tempo (about 600 ms) but vanishes when the difference between the tempo of the external rhythm and the individual's movement tempo is too high [45,87,88]. Furthermore, our results also suggest that a tempo quicker than the spontaneous tempo is detrimental for general learning. As in the IrregAud condition, the deleterious effect of the auditory metronome in the FastRhyth condition could be explained by distractor effect.

## Specific learning is enhanced with congruent audio-visual stimulations and regular rhythmic auditory stimulations

Specific learning of the sequence is attested by an increase in RTmean and errors at B6 (sequenced visual stimulations) compared to B5 (random visual stimulations). This means that a larger increase in RTmean and errors between B5 and B6 highlights a larger specific learning of the visuo-motor sequence. Our results indicate that RTmean increased for all groups, hence suggesting that each condition led to a specific learning of the visuo-motor sequence. However, errors increased more in the conditions with congruent audio-visual stimulations (CongrAV) and regular rhythmic auditory stimulations (RegAud) than in control conditions. Even though the number of participants is small, the relatively high $BF_{10}$ means that the observed effect is real. Even if we expected to find the effects on RTmean rather than on errors, this result is in accordance with our hypotheses. Hence, both the congruent audio-visual stimulations (CongrAV) and the regular rhythmic auditory stimulations (RegAud) enhance procedural learning of the sequence. Benefits in the CongrAV and RegAud conditions are not due to the introduction of two sources of stimulation rather than one source of stimulation, given than the IrregAud et FastRhyth, which provide two sources of stimulation, did not enhance learning.

Our result in the CongrAV condition is in line with the findings of [43,89,90] showing that practice with audio-visual stimulations would improve the acquisition of visual motion-detection skills faster compared to practice with visual stimulations only. The advantage of audio-visual stimulations compared to visual stimulations could be attributed to several processes related to multisensory integration such as (1) a faster detection of audio-visual stimulations than visual stimulations [35,39,42], (2) an improvement of spatial attention [91] and (3) a faster visual learning with multisensory stimulations compared to visual stimulations only [43]. Our results also contribute to the debate in the literature distinguishing (1) a modality-specific mechanism proposing that learning occurs in each modality separately and (2) a modality-general mechanism in which learning is independent across modalities [90,92]. In line with previous results [90], our results tend to be in favor of the latter purpose. Indeed, in [90] authors showed that learners are able to extract statistical regularities from audiovisual input and to integrate it into audio and visual streams separately. In our case, even if the auditory stimulations alone don't provide any cue regarding the sequence, it seems that they still helped the learning of the visual sequence when they were presented simultaneously with the visual stimulations. Therefore, our results are in line with the purpose that procedural learning is sensitive to multimodal input.

As regard to the benefice of regular rhythmic auditory stimulations (RegAud), our results on errors is surprising given that previous studies in the literature indicate that RegAud

quicken RTmean compared to visual stimulations [57]. Our results suggest that RegAud improve learning of the motor sequence by modulating errors (i.e., response on a wrong spatial location), suggesting that RegAud enhance spatial encoding of the motor sequence. This facilitation is in line with previous results showing an improvement of movement stability in both time and space with RegAud [48–50]. Interestingly we found facilitation in the RegAud condition even if we did not manipulate directly the temporal pattern of the to-be-learned material as it was done in most previous studies [see for example [32,93,94]. Indeed, the sequence of position was played out through the visual modality whereas we implemented a temporal structure through auditory stimuli. One hypothesis is that the temporal regularity of these stimuli could have prepared the attentional system to deal with specific stimuli arriving in the same temporal pattern [e.g., [95,96]. Indeed, these effects have already been shown using other tasks of implicit learning of pitch structures [97,98], working memory [99], and statistical learning of artificial language [100].

Moreover, this tempo seems to be well suited given that the optimum tempo for motor synchronization is between 400 and 800ms [101]. Overall, the regularity of the RegAud may have facilitated the learning of the visual stimulations sequence and the increase in the number of errors at the random block (B6) suggests that participants continue to inappropriately play the sequence out [18].

Note that we used an ambitious design with six different conditions. However, all of them were required to understand the overall effect. For example, we showed that it is not only the regularity of the tempo that is decisive but that this tempo is in adequacy with the motor task to facilitate the learning. Moreover, the small sample might have led to underestimate the effects. However, a within-participants design would not have been possible because of (1) the possible transfer or interference effects between conditions and (2) the length of the experiment (6 different conditions x 6 blocks of 100 stimuli + attentional tests). Despite this ambitious design, we found some promising results that need to be explored more deeply and replicated.

## Conclusion

For the first time, our results provide a strong argument in favor of the benefits of audio-visual and regular rhythmic auditory stimulations on procedural learning. This benefit was absent in the control conditions. Given that the addition of auditory information does not automatically enhance procedural learning (control conditions), the benefits cannot be attributed to the addition of auditory information but actually to the rhythmic structure of the auditory stimulations and to the temporal congruency of the auditory and visual stimulations. It suggests that regular rhythmic audio-visual stimulations seem to be a relevant condition to improve procedural learning of perceptual-motor sequences. Even if these preliminary results need for replications and extension with a retention test (with reintroduction of the repeated sequence in another Block, B7), future research is required to find out how sequence learning and temporal information are precisely related, possibly with investigations of the temporal structure of the sequence and the cerebral correlates of procedural learning with rhythmic multisensory stimulations.

## Supporting information

**S1 Table. Bayes factors levels.** Table for the interpretation of each Bayes factors level. Adapted from Jeffreys (1998).
(DOCX)

**S1 File. SRTT data.**
(CSV)

**S2 File. Auditory sounds—Regular auditory sequence.**
(WAV)

**S3 File. Auditory sounds—Irregular auditory sequence.**
(WAV)

## Acknowledgments

We would like to thank Catalina Onofrei for her helpful revision of English and Robert French for his advices in Bayesian statistics. We also would like to thank Manuel Mercier for his help in programming the experiment.

## Author Contributions

**Conceptualization:** Yannick Lagarrigue, Céline Cappe, Jessica Tallet.

**Formal analysis:** Yannick Lagarrigue.

**Investigation:** Yannick Lagarrigue.

**Methodology:** Jessica Tallet.

**Software:** Jessica Tallet.

**Supervision:** Jessica Tallet.

**Validation:** Céline Cappe, Jessica Tallet.

**Writing – original draft:** Yannick Lagarrigue, Jessica Tallet.

**Writing – review & editing:** Yannick Lagarrigue, Céline Cappe, Jessica Tallet.

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
