## [Decision Letter · Decision Letter 0]

10 May 2021

PONE-D-21-00943

Rhythmic and audio-visual stimulations enhance procedural learning of a perceptual-motor sequence in healthy adults: a pilot study

PLOS ONE

Dear Dr. Lagarrigue,

Thank you for submitting your manuscript to PLOS ONE, and we wish to apologize again about the unusually long delay in returning to you the reviewer comments. Both reviewers (and I) were generally positive about the contribution, but both raise substantial issues and questions that need to be addressed before the manuscript fully meets PLOS ONE’s publication criteria. Therefore, we invite you to submit a revised version of the manuscript that addresses the points raised during the review process.

Hopefully the distinction is clear between what is suggested or recommended (e.g. "...Figure 2 could...") versus what is required to address (any direct questions), but feel free to reach out with any questions about that.

In addition to the scientific concerns, I would like to point out that the Data Availability requirement has not been met for this submission. Specifically, the data availability questions state: "Stating ‘data available on request

from the author’ is not sufficient. If your data are only available upon request, select ‘No’ for

the first question [Data Availability] and explain your exceptional situation in the text box." In most cases, PLOS ONE policy requires data to be deposited in a public repository or submitted as supplementary information.

We look forward to receiving your revised manuscript.

Kind regards,

Christopher R. Fetsch

Academic Editor

PLOS ONE

Journal Requirements:

2. Please ensure that you have provided sufficient detail on participant recruitment in the Methods section.

3. Thank you for including your ethics statement:  "All participants provided informed consent prior to data collection. Procedure was in accordance with the Declaration of Helsinki and followed institutional ethics board guidelines for research on humans (IRB00003888)".   

3.1. Please amend your current ethics statement to include the full name of the ethics committee/institutional review board(s) that approved your specific study.

3.2. Please amend your current ethics statement to confirm that your named institutional review board or ethics committee specifically approved this study.

4.3. Please provide additional details regarding participant consent. In the ethics statement in the Methods and online submission information, please ensure that you have specified what type you obtained (for instance, written or verbal, and if verbal, how it was documented and witnessed). If your study included minors, state whether you obtained consent from parents or guardians. If the need for consent was waived by the ethics committee, please include this information.

Reviewers' comments:

Reviewer's Responses to Questions

**Comments to the Author**

1. Is the manuscript technically sound, and do the data support the conclusions?

Reviewer #1: Partly

Reviewer #2: Yes

2. Has the statistical analysis been performed appropriately and rigorously? 

Reviewer #1: Yes

Reviewer #2: Yes

3. Have the authors made all data underlying the findings in their manuscript fully available?

Reviewer #1: No

Reviewer #2: Yes

4. Is the manuscript presented in an intelligible fashion and written in standard English?

Reviewer #1: Yes

Reviewer #2: Yes

5. Review Comments to the Author

Reviewer #1: The authors investigate the effects of auditory stimulations on learning a visuomotor sequence (serial reaction time task). Their findings suggest that, when these stimulations are congruent with the task and/or rhythmic at a regular tempo, they promote procedural learning. Their findings are interesting, and indeed show that auditory stimulations affect procedural learning of a motor sequence. I have some comments/questions as to whether this effect amounts to enhancement; my reservations may be resolved with some additional clarifications.

1) The evidence for enhancement comes from studying specific learning (comparing performance in late learning of a sequence vs. subsequent performance in a random sequence block). The reaction times are similar across groups, but the error rates are different, which is the evidence for enhancement in the RAS and CAV groups.

While SRTT studies seem to mostly rely on RT analysis, I agree with the authors that the error rate is also important and an indication of learning. However, one issue with looking at error rates (and their differences) is that they might be more likely to have floor/ceiling effects – for example, a participant might have little in terms of errors in the random block (B6), thus can't be much better in B5. In particular, maybe the VO group (essentially the “vanilla” SRTT group, against which we compare for enhancement) is simply pretty good in both B5 and B6, with little errors.

So, given the importance of errors in justifying the paper’s thesis – that there is enhancement of learning – it will be helpful to see the absolute error rates during each block (like Figure 2, but for errors), rather than just the difference in error rates between B5 and B6.

2) What could obscure this enhancement when it comes to general learning? If we assume general learning encompasses specific learning (assessed by comparing B5 with B6) among other things, one would expect that increases in specific learning would increase general learning as well.

3) Another limitation in comparing learning across the different groups is that they start from different baselines: e.g. there are large differences in reaction times in B1, before much learning is supposed to occur. In other words, there might be a confound due to differences in baseline performance.

4) As a minor point – auditory stimuli might also act as cues to retrieve the previously learned sequence. So, during the random block, auditory stimuli might promote retrieval of the sequence (even though it is not helpful at that block) – in that way, we could see an over-expression of the previously learned sequence, which would not necessarily mean that the sequence was learned to a greater extent previously.

5) Participants might be learning the sequence but also learning to deal with the auditory stimulation – it might be helpful to show that their performance was stable *before* the sequence was introduced.

Errors and requests for clarification:

1) The caption in Figure 2 does not match the legend in terms of describing the different curves (*for the purpose of my review, I assumed the legend is correct*): it mentions RAS as yellow circles, whereas the legend shows them as purple squares; QRAS as gray circles, whereas the legend shows them as yellow triangles, etc. Please make sure all the colors/symbols mentioned in the caption are correct. The caption also mentions * / # that are not shown in the figure. Captions for figures 3a/3b also mention asterisks that are not shown.

2) p.15 , second to last line: “a larger improvement of errors” – is it meant “a larger increase of errors”?

3) Clarification: Page 4 “… than with irregular metrics (i.e. isochronous intervals)” - Aren’t isochronous intervals also regular tempo?

4) Having “results” as a subtitle for the methods section (2.6) might be not helpful since this is also section 3

5) Typos on Page 16, bottom: one of these two variable*s* decrease*s*… the performance increase*s*…. The IAS condition did not led -> lead

Reviewer #2: The present study investigates the potential benefit of additional auditory stimulation (with temporal regularity or not) on visual sequence learning, as implemented by a classical SRTT. The study is timely and novel, proposing to combine temporal regularity processing with prediction and sequence learning. The project is very interesting and promising. However, the design with 6 between-participants conditions and only 10 participants per condition is ambitious and might lead to underestimate the effects or the missing observation of differences for RTs. The manuscript is missing numerous details and requires some explanations and clarifications (see below). Some streamlining of the writing and section presentation should also help the reader.

The manuscript could also benefit from integrating the present work into other previous research. Numerous statistical learning studies investigating the influence of concurrent or implemented temporal structures are missing in the introduction and discussion (e.g., Buchner & Steffens, 2001; Shin & Ivry 2002; Selchenkova 2014a,b). The paper might also benefit from the integration of the Dynamic Attending Theory and its hypothesis of metric binding (see for example Jones, 2016, 2019). Further relevant references for the present work are the papers by Fujii & Wan (2014), Patel & Iversen (2014) and Tierney & Kraus (2014).

Buchner, A., and Steffens, M. C. (2001). Simultaneous learning of different regularities in sequence learning tasks: limits and characteristics. Psy- chol. Res. 65, 71–80.

Fujii, S., & Wan, C. Y. (2014). The role of rhythm in speech and language rehabilitation: The SEP hypothesis. Frontiers in Human Neuroscience, 8.

Jones, M. R. (2016). Musical time. In S. Hallam, I. Cross, & M. Thaut, The Oxford Handbook of Music Psychology (2nd ed.). Oxford University Press.

Jones, M. R. (2019). Time will tell. Oxford University Press.

Patel, A. D., & Iversen, J. R. (2014). The evolutionary neuroscience of musical beat perception: The Action Simulation for Auditory Prediction (ASAP) hypothesis. Frontiers in Systems Neuroscience, 8.

Selchenkova et al (2014a). Metrical presentation boosts implicit learning of pitch structures. PLOS ONE 9(11): e112233

Selchenkova et al. (2014b). The influence of temporal regularities on the implicit learning of pitch structures. Quarterly Journal of Experimental Psychology, 67, 2360–2380.

Shin, J. C., & Ivry, R. B. (2002). Con- current learning of temporal and spatial sequences. J. Exp. Psychol. Learn. Mem. Cogn. 28, 445–457.

Tierney, A., & Kraus, N. (2014). Auditory-motor entrainment and phonological skills: Precise auditory timing hypothesis (PATH). Frontiers in Human Neuroscience, 8.

Methods

- For the musical background of the participants, it would be relevant to further add the information of musical training in terms of average and SD. Table 1 does not present the participant characteristics (see page 5). One table seems to be missing and the reference to table 1 related to the pre-test condition should read table 2?

- Response-Stimulus-Interval was 200ms. In absence of response, the highlighted square remained on the screen for 3000ms. Did the sequence then go on with the next highlighted square? Please clarify.

- Please clarify the construction of B6. Does “same pseudo-random sequence following the same rules” now refer to all 100 stimuli? Or also a repeating sequence of 10 cycling through ten times?

- Considering the discussions in SRTT research about what is learned during exposure, why didn’t the authors use two controlled Second-Order-Conditional sequences (see research by Cleeremans or Destrebecqz), counterbalancing as test or training across participants?

- Why was the first block referred to as B0? What was the difference between B0 and B1? I guess B0 was presented without auditory stimulation, please clarify.

- Page 7 “This sequence was randomly assigned to each participant” – Does this refer to one of the four sequences (A, B, C or D)? Why were these sequences not respectively used as the test-block B6 sequences (such as, for example, Participant 1 would get sequence A for blocks B0 to B5 and sequence B for block 6 while Participant 2 would get the reverse attribution)?

- Why did the authors not include the expected B7 block that returns back to the exposure sequence (B0 to B5), as usually done in SRTT?

- The material presentation requires further details: what was the irregular pattern used in IAS? Was it the same for all blocks? If yes, participants might have learned it and it might have become less unexpected over time.

- Was the auditory material presented during all blocks (B0 to B6)?

- How was the sound of the auditory stimulation made? At which pitch height and loudness level was it presented? How was it presented (via headphones or free field)?

- What was told to the participants about the purpose of the sound?

- When were the TAP tests presented?

- For the explicit knowledge testing, why the choice or 3-alternatives rather than a subjective scale with more gradings, which might be more sensitive to differences?

Results:

- Were RTs only kept from correct responses? It is not clear how the average and the variability was calculated: “across Blocks of trials were computed” Wouldn’t it need to be done across trials of each of the blocks?

- Figure 2 could integrate an extra data point presenting RTs at B0 (without connecting lines to B1).

- Should title 2.6. read “data analyses”? This information could be combined with the data analysis section further up, allowing for streamlining the manuscript and removing redundancies.

- 2.6.1: The correlations should also be completed with the task learning score B0-B5 for attention and B0 analyses.

- 2.6.2: the two ANOVAs are redundant – using blocks as additional factor (to investigate the difference between B5 and B6) or running the analysis on the computed difference should provide equivalent results (as shown by same F-values, p-values and partial eta2, see page 13).

- Page 11: Please present the used categories of the Bayes factor interpretation in the text.

- 3. Results:

o Page 12: add exact p-values for the correlations into the text (or the table). Were Bayesian analyses also run for these pre-test analyses (as for 3.2)?

o Page 13: why were additional Friedman tests ran for IAS?

o Figure 2 suggests that the RTs differ in absolute terms between at least a subset of the conditions (e.g., fastest RT for RAS, followed by VO, QRAS and IAS being slowest). It might be interesting to normalize response times (e.g., using z-scores for each participant). This would remove these inter-group differences in terms of absolute response speed, but might allow for being more sensitive to reveal differences in general and specific learning.

Discussion

- Page 18/19: The discussion of the QRAS condition is interesting, but it requires that the overall speed of response of the participants in this condition is not faster (as faster RTs would leave less room for improvement, that is decrease of RTs)

- Page 19: This study showed some learning on errors but not on RTs. Was RTs particularly fast in comparison to other studies? The explanation that error modulation with RAS reflects enhanced spatial encoding does not seem to be complete because one could also argue that enhanced spatial encoding of the motor sequence should lead to modulation of RTs.

The manuscript could gain in clarity and should be checked by a native English speaker too. At some points, clarity in writing could be increased (e.g., “Attentional performance was assessed to explore the link between visuo-motor learning and attentional skills. They were also used to make sure that groups …”) I guess ‘they’ refers to the two attentional tests, but they were not mentioned here.

The abbreviations used for the six conditions are somewhat abstract and difficult to remember; replacing them by more informative ones would facilitate reading (e.g., VisOnly, CongrAV, NonCongrAV, RythAud, IrregAud, FastRyth). Also, “rhythmic” does not seem to be the best wording for the “rhythmic auditory stimulation”, in particular with its contrast to the Irregular auditory stimulation, the label “Regular auditory stimulation” (RegAud) seems more appropriate.

6. PLOS authors have the option to publish the peer review history of their article (what does this mean?). If published, this will include your full peer review and any attached files.

Reviewer #1: No

Reviewer #2: No

---

## [Author Response · Author response to Decision Letter 0]

21 Jul 2021

Dear Professor Fetsch and reviewers, 

We would like to thank you for reviewing our manuscript entitled “Rhythmic and audio-visual stimulations enhance procedural learning of a perceptual-motor sequence in healthy adults: a pilot study” for publication in PLOS ONE. All of you did very insightful comments and proposed relevant changes that highly improved the quality of the manuscript. 

The manuscript has been revised for better readability according to the suggestions. We responded to all of the remarks with associated changes highlighted in blue. See all changes in the document "Response to Reviewers".

In accordance with PLOS ONE’s policy, we uploaded anonymized data set as Supporting Information files and we adjusted titles and Figures’ style to meet PLOS ONE’s requirements. 

All authors have reviewed and agreed to the submission of the revised manuscript. We hope that our responses fix all the issues that you raised and that the manuscript is now acceptable for publication. Please do not hesitate to contact me if there are any questions.

---

## [Decision Letter · Decision Letter 1]

17 Aug 2021

PONE-D-21-00943R1

Rhythmic and audio-visual stimulations enhance procedural learning of a perceptual-motor sequence in healthy adults: a pilot study

PLOS ONE

Dear Dr. Lagarrigue,

Thank you for submitting your manuscript to PLOS ONE. After careful consideration, we feel that it largely meets PLOS ONE’s publication criteria, although there are a small number of remaining issues to address and suggestions for improvement. Therefore, we invite you to submit a revised version of the manuscript that addresses these issues.

Both reviewers and I appreciate the effort in response to the previous comments, and all agree the resulting manuscript is greatly improved. Reviewer 2 listed a number of relatively minor concerns/suggestions, and one or two more substantial ones, that should be addressed. 

We look forward to receiving your revised manuscript.

Kind regards,

Christopher R. Fetsch

Academic Editor

PLOS ONE

Journal Requirements:

Reviewers' comments:

Reviewer's Responses to Questions

**Comments to the Author**

1. If the authors have adequately addressed your comments raised in a previous round of review and you feel that this manuscript is now acceptable for publication, you may indicate that here to bypass the “Comments to the Author” section, enter your conflict of interest statement in the “Confidential to Editor” section, and submit your "Accept" recommendation.

Reviewer #1: All comments have been addressed

Reviewer #2: (No Response)

2. Is the manuscript technically sound, and do the data support the conclusions?

Reviewer #1: Yes

Reviewer #2: Partly

3. Has the statistical analysis been performed appropriately and rigorously? 

Reviewer #1: Yes

Reviewer #2: No

4. Have the authors made all data underlying the findings in their manuscript fully available?

Reviewer #1: Yes

Reviewer #2: Yes

5. Is the manuscript presented in an intelligible fashion and written in standard English?

Reviewer #1: Yes

Reviewer #2: Yes

6. Review Comments to the Author

Reviewer #1: I’d like to thank the authors for their response. I believe the inclusion of Figure 4a, and block b0 in multiple figures, paint a more clear picture of what’s going on.

Moreover, thank you for the general vs. specific learning clarification. I originally understood general learning as “total” learning, i.e. something that would encompass specific learning; here however general and specific learning are distinct. My one remaining suggestion is that you add a short definition earlier in the manuscript, when these terms first appear, as not all readers might be familiar with the distinction.

-

Reviewer #2: The revised manuscript is considerably improved in clarity and presentation. I still have the following points that the authors should address in a revision.

- The manuscript should address the concern of the ambitious design of six between-participants conditions and only 10 participants per condition, which might lead to underestimate the effects, for example.

- The response letter explains that B0 was another sequence different from the to-be-learned sequence and supposed to act as baseline. However, this would require to be presented without auditory stimulation. If B0 is presented with the same auditory condition as B1, then its purpose is not clear. Indeed, then even B0 is submitted to potential influences of the experimental condition. Using the first block to check that participants did not differ between groups would have served the same purpose. Considering that within-block learning can occur, presenting two different sequences at the beginning might also affect learning (i.e., learning and re-learning) and raises concerns regarding the similarity between B0 and the experimental sequence, which might affect learning differently.

- Page 84 (pdf) “following the previous rules applied to 100 trials” (p. 8) This “citation” is not in the manuscript, please check all citations to make sure that the manuscript contains the same text as claimed in the response letter to the reviewers.

- Page 84 (pdf) “applied to 100 trials” “trials” is a terminology that might lead to confusion, “items” seems more appropriate (here and elsewhere) or clarify. Keep the same wording throughout (sometimes referred to as “stimulations”)

- The suggestion to use the same sequences across participants as test or learning was not to investigate “transfer effects” (page 85 of the pdf), but to control for sequence specific features. Please clarify and integrate your explanation and justification also in the manuscript as other readers might wonder the same.

- Page 53 pdf/page8 manuscript: “a last Block (B6) presented the visual stimulations in a pseudo-random fashion”. The authors should spell out the exact sequence used (as they do for sequences A, B, C and D) so that the reader can compare it with the different training sequences used.

- Page 86 pdf: The temporal pattern consisted of 160 stimuli. This needs to be clarified as one block consisted of 100 items. In addition, the authors should explain how the “intervals between them were determined randomly”, what were the possible intervals (range min/max, sampling etc]. Also provide the used sequence as supplementary material in written form and audio file.

- It would be helpful for the readers to appreciate the work done by the authors and the manipulations used by adding as supplementary materials some short video excerpts displaying the paradigm and illustrating the auditory congruency or not (e.g., allowing for evaluating also the potential disturbance of the irregular sequence).

- page 87. The presentation at 80dB seems quite loud. Why did the authors opt for this loudness level?

- Page 89 (pdf): I agree with the authors that Jeffreys table could be placed in an appendix section.

- Page 90: Thanks for this response. The authors should present their normalised RT data also into the manuscript, as they propose here. The z-score transformation (pooled across blocks) should remove the between-group speed differences and allow for the discussion (previous version page 18/19 of the manuscript). However, I do not see the purpose to use z-scores standardized by block as this does not allow for testing learning effects (see bottom part of page 90).

- page 10 manuscript: “fixed to randomised” - pseudorandomized ?

- page 19 manuscript : more difficult than rhythmic auditory stimulations” - does this here refer to “regular” rhythmic stimulations? Please clarify (and check throughout the manuscript that the labelling is clear).

- The authors should also extend their discussion to the influence of multi-dimensional or dual cues in learning (e.g., Mitchel & Weiss, 2011 JEP:LMC).

7. PLOS authors have the option to publish the peer review history of their article (what does this mean?). If published, this will include your full peer review and any attached files.

Reviewer #1: No

Reviewer #2: No

---

## [Author Response · Author response to Decision Letter 1]

27 Sep 2021

Dear Professor Fetsch and reviewers, 

We would like to thank you for your positive and constructive comments on our manuscript entitled “Regular rhythmic and audio-visual stimulations enhance procedural learning of a perceptual-motor sequence in healthy adults: a pilot study”. 

The manuscript has been revised for better readability according to the suggestions. We responded to all of the suggestions with associated changes highlighted in blue. 

All authors have reviewed and agreed to the submission of the revised manuscript. We hope that our responses fix all the concerns that you raised and that the manuscript is now acceptable for publication. Please do not hesitate to contact me if there are any questions.

---

## [Editor Report · Decision Letter 2]

13 Oct 2021

Regular rhythmic and audio-visual stimulations enhance procedural learning of a perceptual-motor sequence in healthy adults: a pilot study

PONE-D-21-00943R2

Dear Dr. Lagarrigue,

We’re pleased to inform you that your manuscript has been judged scientifically suitable for publication and will be formally accepted for publication once it meets all outstanding technical requirements.

Kind regards,

Christopher R. Fetsch

Academic Editor

PLOS ONE
---

## [Editor Report · Acceptance letter]

26 Oct 2021

PONE-D-21-00943R2 

Regular rhythmic and audio-visual stimulations enhance procedural learning of a perceptual-motor sequence in healthy adults: a pilot study 

Dear Dr. Lagarrigue:

I'm pleased to inform you that your manuscript has been deemed suitable for publication in PLOS ONE. Congratulations! Your manuscript is now with our production department. 

Kind regards, 

on behalf of

Dr. Christopher R. Fetsch 

Academic Editor

PLOS ONE